# The gut microbiota is a transmissible determinant of skeletal maturation

Abdul Malik Tyagi[1,2], Trevor M Darby[2,3], Emory Hsu[1,2], Mingcan Yu[1,2], Subhashis Pal[1,2], Hamid Dar[1,2], Jau-Yi Li[1,2], Jonathan Adams[1,2], Rheinallt M Jones[2,3], Roberto Pacifici[1,2,4]*

[1]Division of Endocrinology, Metabolism and Lipids, Department of Medicine, Emory University, Atlanta, United States; [2]Emory Microbiome Research Center, Emory University, Atlanta, United States; [3]Department of Pediatrics, Emory University, Atlanta, United States; [4]Immunology and Molecular Pathogenesis Program, Emory University, Atlanta, United States

**Abstract** Genetic factors account for the majority of the variance of human bone mass, but the contribution of non-genetic factors remains largely unknown. By utilizing maternal/offspring transmission, cohabitation, or fecal material transplantation (FMT) studies, we investigated the influence of the gut microbiome on skeletal maturation. We show that the gut microbiome is a communicable regulator of bone structure and turnover in mice. In addition, we found that the acquisition of a specific bacterial strain, segmented filamentous bacteria (SFB), a gut microbe that induces intestinal Th17 cell expansion, was sufficient to negatively impact skeletal maturation. These findings have significant translational implications, as the identification of methods or timing of microbiome transfer may lead to the development of bacteriotherapeutic interventions to optimize skeletal maturation in humans. Moreover, the transfer of SFB-like microbes capable of triggering the expansion of human Th17 cells during therapeutic FMT procedures could lead to significant bone loss in fecal material recipients.

*For correspondence: roberto.pacifici@emory.edu

**Competing interests:** The authors declare that no competing interests exist.

## Introduction

Osteoporosis has a devastating impact on quality of life and is a significant cause of disability and morbidity in both women and men. Low bone mineral density (BMD), a key feature of osteoporosis, results from suboptimal skeletal maturation culminating in low peak bone density by 30 years of age, and/or by accelerated bone loss thereafter. Interestingly, there is considerable natural variation in peak BMD within populations, with most of the variation being attributed to genetic heterogeneity (*Boudin et al., 2016*; *Peacock et al., 2002*; *Estrada et al., 2012*). However, the remaining non-genomic factors that contribute to peak BMD variance remain enigmatic. A well-recognized determinant of phenotypic variability within populations is the composition and community structure of the gut microbiome (*Sommer and Bäckhed, 2013*; *Tremaroli and Bäckhed, 2012*). Indeed, studies undertaken in both humans and animals have pointed to the gut microbiome as a potential regulator of skeletal maturation. Among them are the reports that mice raised in germ-free (GF) conditions have altered BMD, bone volume and bone structure compared to conventionally raised (Conv.R) mice (*Sjögren et al., 2012*; *Novince et al., 2017*; *Schwarzer et al., 2016*; *Yan et al., 2016*). Additional evidence showing that the microbiome is a significant regulatory factor for skeletal health include reports that the gut microbiome affects the bone waste induced by estrogen deficiency (*Li et al., 2016*), glucocorticoids (*Schepper et al., 2020*) and by diabetes (*Zhang et al., 2015*). Furthermore, we showed that the presence of the gut microbiota is required for parathyroid hormone (PTH) to stimulate both bone formation and bone resorption (*Yu et al., 2020*; *Li et al., 2020*). It is also known that antibiotic treatment which depletes the gut microbiome increases bone

density in mice (*Yan et al., 2016*), while bacterial recolonization post-antibiotic treatment results in bone loss (*Schepper et al., 2019*). Moreover, variations in the composition of the microbiome affect not only BMD, but also bone strength by its actions on bone material properties (*Guss et al., 2017*).

Humans are initially colonized by microbes at birth, with the microbiota eventually reaching maturity at about 3 years of age (*Yatsunenko et al., 2012*; *Rodríguez et al., 2015*; *Kashtanova et al., 2016*). Thereafter, gut bacterial species are thought to adapt within individual microbiomes enabling their long-term colonization into adulthood (*Zhao et al., 2019*). Early microbial colonization and microbiota succession is influenced by the maternal microbiota, although a precise measure of the extent to which a mother microbiome is acquired by the offspring has been challenging to ascertain (*Benson et al., 2010*; *McKnite et al., 2012*). Indeed, strict vertical inheritance of the gut microbiota is thought not to occur in the human microbiome (*Groussin et al., 2020*), although recent studies using deep metagenomic microbiome sequencing have reported that in most cases the mother's dominant strains were transmitted to her child (*Yassour et al., 2018*). Therefore, early gut colonizers, if retained, do have the potential to exert lasting physiological effects on health and disease for most or perhaps all of adult life (*Faith et al., 2013*; *Faith et al., 2015*). It is also noteworthy that long lasting modifications to the mature microbiota require substantial interventions, including permanent dietary changes, major changes in the health status of the host, or extensive microbial manipulations such as a long-term course of antibiotics (*Lozupone et al., 2012*). Although microbiome sequencing approaches have shown a mother's dominant strains were transferred to her child (*Yassour et al., 2018*), few studies have shown that phenotypes elicited by dominant strains are also transferred to offspring. In addition, few studies have demonstrated that phenotypes elicited by strains within microbiomes are transferred by co-habitation. Discovering that phenotypes that have a negative impact on bone health are transferred within the microbiome would also identify a window of opportunity for intervention with the ultimate goal of promoting the optimal skeletal development.

An example of a gut microbe that elicits a salient and tractable phenotype is segmented filamentous bacteria (SFB), which are spore-forming, Gram-positive commensal bacteria. In the mouse, SFB drive the development and expansion of Th17 cells that are produced and reside in the intestinal lamina propria (*Ivanov et al., 2009*; *Ivanov et al., 2008*; *Gaboriau-Routhiau et al., 2009*). Th17 cells are an osteoclastogenic population of CD4+ T cells (*Sato et al., 2006*; *Miossec et al., 2009*) defined by their capacity to produce IL-17 (*Basu et al., 2013*). The presence of SFB in the intestine of healthy mice leads to lower bone density (*Hathaway-Schrader et al., 2020*), which occurs via SFB-driven expansion of Th17 cells in the gut, migration of Th17 cells to the bone marrow (BM), and increased secretion of IL-17 in the BM (*Yu et al., 2020*; *Li et al., 2020*). These observations are a well-characterized mechanism of how a gut microbe can act as a modulator of a host phenotype, in this case the skeletal response. These observations also offer a powerful model to establish proof of principle that phenotypes are transmitted within the microbiome. Pertaining to relevance to public health, in humans, about twenty non-virulent gut bacterial strains are known to induce Th17 cell differentiation (*Atarashi et al., 2015*; *Tan et al., 2016*). If any of the human microbiome-residing Th17 cell-inducing bacteria are transmitted from a mother to her child, then they may turn out to be a significant influencer of, or an impediment to the child's skeletal development. Furthermore, the approach of fecal microbiome transplantation (FMT) from healthy donors to patient with diseases such as p*seudomembranous colitis* of the large intestine due to an overgrowth of Clostridioides is increasingly practiced. The transfer *of* gut microbes that expand Th17 cells in the gut of recipients during FMT may lead to lower bone density, particularly in the context of hyperparathyroidism (*Yu et al., 2020*). These observations underscore an exigent need to discover if bone phenotypes are communicable with the transfer of gut microbes between individuals.

With this goal, in this study we assessed the role of gut microbiota as a transmissible influencer of skeletal development and homeostasis. Using gnotobiotic murine models of early microbiome establishment through maternal contacts or cohabitation, we show that skeletal phenotypes are transferable within the fecal microbiome. We also show that the transfer from the mother to offspring of microbes that induce Th17 cell expansion in the gut is a significant negative determinant of bone health structure later in life. Together, our data supports the notion that the gut microbiome acts as a factor that can be transmitted between individuals and elicit significant effects on skeletal development. These findings also identify the acquisition of the microbiome as a window of opportunity for therapeutic interventions to overcome gut microbe-orchestrated suboptimal skeletal maturation.

## Results

### Maternal microbiome regulates skeletal maturation

To investigate the extent to which maternal microbiome affects the skeleton of the offspring we made use of two strains of mice, C3H/HeN and C57BL/6 purchased from Taconic biosciences (herein referred to as TAC mice) raised in conventional conditions or GF conditions. C3H/HeN mice have higher indices of bone volume and bone stracture as compared to C57BL/6 mice. Conventional TAC C57BL/6 mice were raised in a vivarium that contains SFB, which are potent microbial inducers of Th17 cell expansion in intestinal tissue (*Ivanov et al., 2009*). Conventional TAC C3H/HeN mice were raised in a SFB⁻ facility. In these experiments, we transferred fecal material of each conventional mouse strain to GF mating pairs of the other mouse strain, i.e. transfer fecal material from a conventioanlly raised C57BL/6 to a GF C3H/HeN breeding pairs, and vice versa. We then assessed bone structure and turnover in 16-week-old female mice of the F1 generation produced by the conventionalized breeding pairs, which were colonized with the mother's microbiome from birth (*Figure 1A*).

Microbiome sequencing confirmed that the bacterial community structure was different in conventional C57BL/6 and C3H/HeN mice (*Figure 1B*). We also confirmed the absence of SFB in GF C57BL/6 mice that were colonized with C3H/HeN microbiome, and the presence of SFB in C3H/HeN germ-free mice that were colonized with C57BL/6 microbiome (*Figure 1C*). Herein, we referred to mice colonized with fecal material from donor of the different genetic background as 'discordant' mice, whereas mice colonized with fecal material from a donor of the same genetic background are referred to as 'concordant' mice. At 16 weeks of age C3H/HeN concordant mice had higher bone volume/total volume ratio (BV/TV) compared to concordant C57BL/6 mice (*Figure 1D*). SFB⁺ mice have a lower BV/TV compared to SFB⁻ mice (*Yu et al., 2020*; *Hathaway-Schrader et al., 2020*). Corroborating the relevance of SFB for skeletal maturation, we showed that the discordant C3H/HeN mice had lower femur BV/TV ratio, trabecular number (Tb.N), and trabecular thickness (Tb.Th), and higher trabecular separation (Tb.Sp) than concordant C3H/HeN mice (*Figure 1D–G*). By contrast, concordant and discordant C57BL/6 mice had similar indices of trabecular volume and structure (*Figure 1D–G*). Moreover, cortical area (Ct.Ar) and cortical thickness (Ct.Th), which are indices of cortical bone structure, were higher in concordant C3H/HeN mice than in concordant C57BL/6 mice, while discordant C3H/HeN mice had lower Ct.Ar and Ct.Th than concordant C3H/HeN mice (*Figure 1H,I*). Analysis of C57BL/6 mice cortical structure revealed that Ct.Ar and Ct.Th were higher in discordant than in concordant mice (*Figure 1H,I*). Together, these findings indicate that TAC C57BL/6 microbiome worsen trabecular and cortical bone maturation. Mechanistic studies revealed that serum CTX, a marker of bone resorption, was higher in concordant C57BL/6 mice than in concordant C3H/HeN mice. Moreover, C3H/HeN microbiome lowered CTX levels in C57BL/6 mice, while C57BL/6 microbiome increased bone resorption in C3H/HeN mice (*Figure 1J*). By contrast, all groups of mice had similar serum levels of osteocalcin, a marker of bone formation (*Figure 1K*).

Since C57BL/6 microbiome is SFB⁺ and C3H/HeN microbiome is SFB⁻, and SFB is a potent inducer of intestinal Th17 cell expansion (*Ivanov et al., 2009*), we assessed the number of Th17 cells in Peyer's Patches (PP) and the BM (*Figure 2—figure supplement 1*). Because the measurement of the absolute number of PP Th17 cells is technically challenging due to variability of the size of the collected PP tissue, PP Th17 cells were quantified as percentage of total CD4⁺ T cells. We confirmed that the number of PP Th17 cells and the levels of small intestine (SI) tissue *Il17a* transcripts were higher in all groups of mice with SFB⁺ microbiome than in those with SFB⁻ microbiome (*Figure 2A, B*). In addition, the number of Th17 cells and the levels of *Il17a* mRNA in the BM were also higher in all mice with an SFB⁺ microbiome, as compared to those with SFB⁻ microbiome (*Figure 2C–E*). IL-17 stimulates the release of further inflammatory cytokines. Accordingly, we found the levels of *Tnfa* mRNA were higher in the BM and SI and of mice with SFB⁺ microbiome (*Figure 2G,H*). BM Receptor activator of nuclear factor kappa-B ligand (*Rankl*), also known as tumor necrosis factor ligand superfamily member 11 (*Tnfsf11*) was higher in mice with SFB⁺ microbiome (*Figure 2I*), although SI *Tnfsf11* mRNA levels were similar in all groups (*Figure 2J*). These data suggest that the signal to expand BM Th17 cells originates in the intestine, and requires Th17 cell-inducing bacteria, such as SFB to be a component of the microbiome. Critically, these data show proof of principle that acquisition of maternal microbiome at birth affects the postnatal skeletal development of the offspring regardless of its genetic strain.

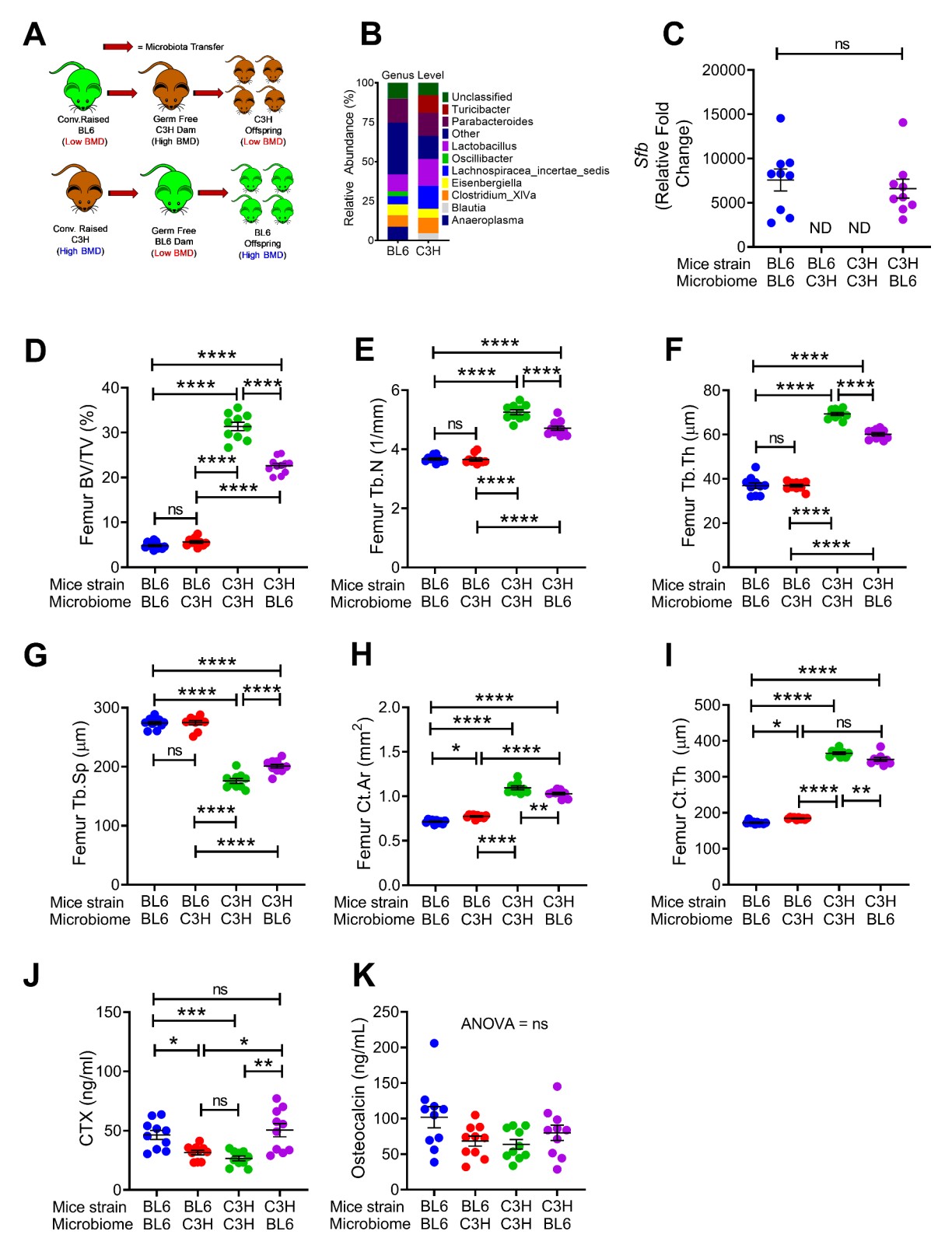

**Figure 1.** Nongenomic effects of gut microbiome on bone volume, structure and turnover are hereditarily transferred. (**A**) Diagram of the experimental outline. Fecal material from conventional C57BL/6 (low BMD) or C3H/HeN (high BMD) mice was transferred to germ-free mating pairs of the other strain. Bone structure and turnover were analyzed in 16-week-old female mice of the F1 generation to determine the influence of the donor microbiota on post-natal skeletal development. (**B**) Fecal microbiome composition in C57BL/6 (BL6) mice and C3H/HeN (C3H) mice. (**C**) Quantitative PCR (qPCR)

*Figure 1 continued on next page*

*Figure 1 continued*

analysis of *Sfb* and total bacterial 16S rRNA genes in fecal samples. (**D**) Femoral trabecular bone volume fraction (BV/TV). (**E**) Trabecular number (Tb.N). (**F**) Trabecular thickness (Tb.Th). (**G**) Trabecular separation (Tb.Sp). (**H**) Cortical Area (Ct.Ar). (**I**) Cortical thickness (Ct.Th). (**J**) Serum levels of CTX, a marker of bone resorption. (**K**) Serum levels of osteocalcin (OCN), marker of bone formation. n = 9–12 mice per group. Data were expressed as mean ± SEM. All data were normally distributed according to the Shapiro-Wilk normality test. Data were analyzed by 2-way ANOVA and post hoc tests applying the Bonferroni correction for multiple comparisons. *$p<0.05$, **$p<0.01$, ***$p<0.001$, ****$p<0.0001$ compared to indicated groups. The online version of this article includes the following source data for figure 1:

**Source data 1.**

## Cohabitation regulated skeletal development

Cohabitation is a key non-genomic factor that influences the development over time of species structure in the microbiota of humans (*Faith et al., 2013*) and mice (*Robertson et al., 2019*). To evaluate the role of cohabitation on skeletal maturation, 10-week-old female C57BL/6 mice purchased from Jackson Laboratory (herein referred to as JAX mice) and C57BL/6 TAC mice were housed separately, or co-housed for 4 weeks (*Figure 3A*). Confirming earlier reports (*Ivanov et al., 2009*), we found JAX and TAC mice to have distinct microbiome composition and diversity (*Figure 3B*). Importantly, while TAC mice were SFB⁺, JAX mice were SFB⁻ (*Figure 3C*), a finding concordant with previous reports (*Farkas et al., 2015*). As expected, co-housing of JAX and TAC mice equalized the composition of the microbiome, and lead to SFB colonization within the JAX mice (*Figure 3B,C*). JAX C57BL/6 mice that were housed alone had a higher femoral BV/TV, Tb.N and Tb.Th and lower Tb.Sp than TAC mice housed alone (*Figure 3D–G*), although they did have similar cortical structure (*Figure 3H,I*). Co-housing of JAX mice with TAC mice resulted in the equalization of all indices of trabecular volume and structure, although it did not affect cortical bone indices (*Figure 3D–I*). TAC mice housed alone had higher levels of serum CTX than JAX mice housed alone (*Figure 3J*). Co-housing of JAX and TAC mice increased the serum levels of CTX in JAX mice, leading to the equalization of CTX levels. By contrast, all groups of mice had similar levels of serum osteocalcin (*Figure 3K*). These findings indicate that post-natal skeletal development is influenced by cohabitation via a microbiome dependent mechanism that affects bone resorption.

We found that TAC mice had a higher number of PP and BM Th17 cells and higher expressions of SI and BM *Il17a* transcripts compared to JAX mice (*Figure 4A–E*). Co-housing resulted in an increase in Th17 cells and *Il17a* mRNA in JAX mice. As a result, at the end of the cohabitation period, Th17 cells and *Il17a* transcript were similar in gut tissues and BM of TAC mice and cohoused JAX mice compared to JAX mice housed alone (*Figure 4A–E*). In addition, TAC mice housed alone had higher expression of SI and BM *Tnfa* than JAX mice housed alone, while co-housing of the two groups increased the levels of *Tnfa* transcripts in JAX mice, up to levels of TAC mice (*Figure 4F,G*). Finally, while SI *Tnfsf11* mRNA levels were similar in all groups (*Figure 4H*), expression of *Tnfsf11* in the BM was higher in TAC mice than in JAX mice housed alone. However, BM *Tnfsf11* mRNA levels increased in JAX mice as a result of co-housing with TAC mice (*Figure 4I*). These data suggest that co-housing of JAX mice with TAC mice induced bone loss in JAX mice by expanding BM Th17 cells and increasing osteoclastogenic cytokine levels.

## Fecal microbiome transfer (FMT) from C57BL/6 TAC mice to antibiotic treated C3H/HeN JAX mice induced bone loss in C3H/HeN TAC mice

Although the gut microbiota in mature animals and humans is resistant to colonization by new organisms, extensive and prolonged microbicidal interventions such as a long-term course of antibiotics may elicit enduring changes (*Yatsunenko et al., 2012*). To investigate the impact on post-natal skeletal development of microbiome changes occurring after the establishment and consolidation of the original microbiome, FMTs were carried out across two strains of mice, and antibiotic treatments were employed. Specifically, groups of 8-week-old female TAC C57BL/6 and TAC C3H/HeN were treated with a cocktail of broad-spectrum antibiotics (Ampicillin; 0.5 g/L, Vancomycin; 0.25 g/L, Neomycin; 0.25 g/L, and Metronidazole; 0.5 g/L) for 2 weeks. 24 hr after the completion of the antibiotic treatment, mice were subjected to oral gavage of a liquid suspension of fecal material from either TAC C57BL/6 mice or TAC C3H/HeN mice (*Figure 5A*). The recipient mice were housed in sterile cages in an isolated environment for 8 weeks before sacrifice (*Figure 5A*). Analysis showed that

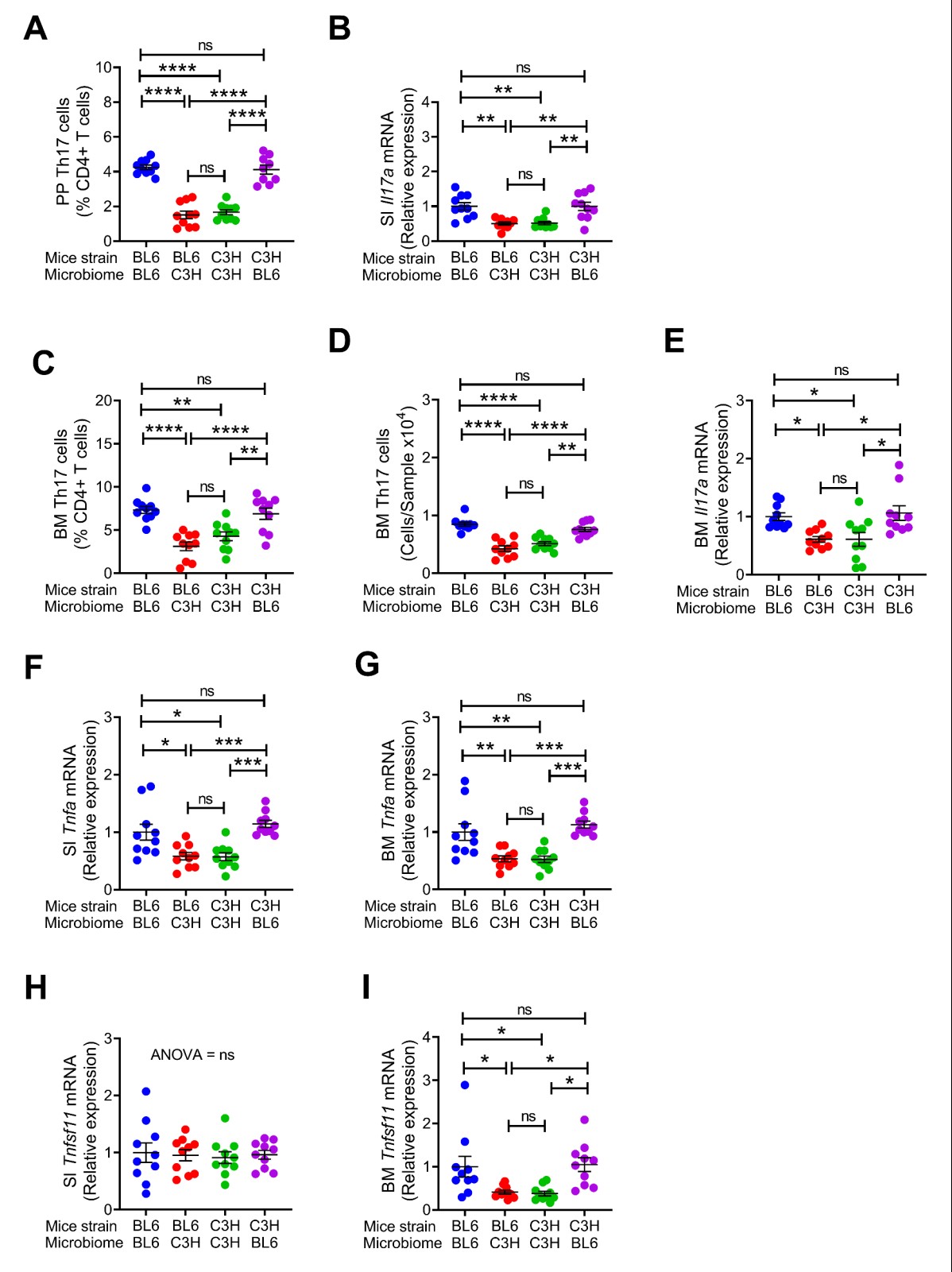

**Figure 2.** Effects of maternal fecal microbime on the frequency of BM and PP Th17 cells and transcript levels of inflammatory cytokines in the offspring. (A) Relative frequency of Th17 cells (IL-17+ CD4+ T cells) in PP. (B) SI *Il17a* mRNA levels. (C,D) Relative and absolute frequency of BM Th17 cells. (E) BM *Il17a* mRNA levels. (F) SI *Tnfa* mRNA levels. (G) BM *Tnfa* mRNA levels. (H) SI *Tnfsf11* mRNA levels. (I) BM *Tnfsf11* mRNA levels. n = 10 mice per group. Data were expressed as mean ± SEM. All data were normally distributed according to the Shapiro-Wilk normality test. Data were analyzed by 2-way

*Figure 2 continued on next page*

*Figure 2 continued*

ANOVA and post hoc tests applying the Bonferroni correction for multiple comparisons. *p<0.05, **p<0.01, ***p<0.001, ****p<0.0001 compared to indicated groups.

The online version of this article includes the following source data and figure supplement(s) for figure 2:

**Source data 1.**

**Figure supplement 1.** Gating strategy used to identify BM (Panel **A**) and Payer's patches (Panel **B**) Th17 cells.

donor C57BL/6 and C3H/HeN mice had distinct microbiome composition and diversity, and that FMT led to the establishment of the donor microbiome in the host mouse (*Figure 5B*). We confirmed that SFB was present in donor C57BL/6 mice, and that antibiotic treated mice subjected to FMT with fecal material from these mice were also SFB$^+$(*Figure 5C*). In-vitro μCT analysis showed that donor C3H/HeN mice had higher femoral BV/TV, Tb.N, Tb.Th and lower Tb.Sp compared to C57BL/6 mice (*Figure 5D–G*). C3H/HeN mice also had higher indices of cortical structure compared to C57BL/6 mice (*Figure 5H,I*). FMT from donor C57BL/6 mice to antibiotic treated C3H/HeN mice altered all indices of trabecular volume and structure in the recipient C3H/HeN mice (*Figure 5D–G*). In addition, FMT from C57BL/6 mice to antibiotic treated C3H/HeN mice induced cortical bone loss in the recipient C3H/HeN mice (*Figure 5H,I*). Conversely, FMT from donor C3H/HeN mice did not affect indices of trabecular and cortical bone in recipient C57BL/6 mice (*Figure 5D–I*). These data suggested that gut microbiota of C57BL/6 mice has modulatory influences on both trabecular and cortical bone structure. In order to investigate the mechanism of bone loss induced by the transfer of C57BL/6 to antibiotic treated C3H/HeN mice, we measured serum levels of markers of bone resorption and bone formation. Analysis showed that serum levels of CTX were significantly higher in antibiotic treated C57BL/6 mice and C3H/HeN mice colonized with C57BL/6 microbiome (*Figure 5J*). However, FMT had no effect on the levels of osteocalcin in all experimental groups (*Figure 5K*). These data indicate that the bone loss that occurred following the transfer of a C57BL/6 microbiome to recipient mice was due to increasing bone resorption.

Additional studies revealed that C57BL/6 donor mice had higher numbers of PP and BM Th17 cells, and higher expression of *Il17a* transcripts compared to C3H/HeN donor mice (*Figure 6A–E*). FMT of fecal material from C57BL/6 donor mice to antibiotic treated to C3H/HeN mice resulted in increased Th17 cell numbers and *Il17a* transcript in recipient mice (*Figure 6A–E*). In addition, antibiotic treated C3H/HeN mice colonized with fecal material from C57BL/6 donor had higher levels of *Tnfa* mRNA in the SI and BM compared to donor C3H/HeN mice (*Figure 6F,G*). While SI *Tnfsf11* mRNA levels were similar in all groups (*Figure 6H*), expression of *Tnfsf11* in the BM was higher in donor C57BL/6 TAC than in C3H/HeN mice. Moreover, BM *Tnfsf11* mRNA levels were increased in antibiotic treated mice colonized with C57BL/6 (*Figure 6I*). These data indicate that the transfer of the C57BL/6 TAC microbiome to an antibiotic treated mouse of a different strain resulted in the induction of an inflammatory environment that led to increased bone resorption and net bone loss. The data also suggest that colonization following a major disruption to the microbiome can change the microbiome composition and induce skeletal phenotypic alterations.

## Discussion

This study demonstrated the contribution of the gut microbiome to skeletal maturation. We showed that the gut microbiome shapes the acquisition of bone volume, structure and turnover when acquired by newborn mice via vertical transfer from the mother or by young mice via co-housing with other mice. Replacement of the original established microbiome with a new microbiome via treatment with broad-spectrum antibiotics followed by FMT, also conditioned skeletal maturation. Transmission of gut microbes, and in particular of SFB, lead to the stimulation of bone resorption in recipient mice of the same, or different genetic strain. Moreover, the gut microbiome was found to act as a major non-genomic heritable factor in the transmission of bone phenotypes, including from mother to offspring. Together, these data showed that the microbiome is a communicable regulator of bone development.

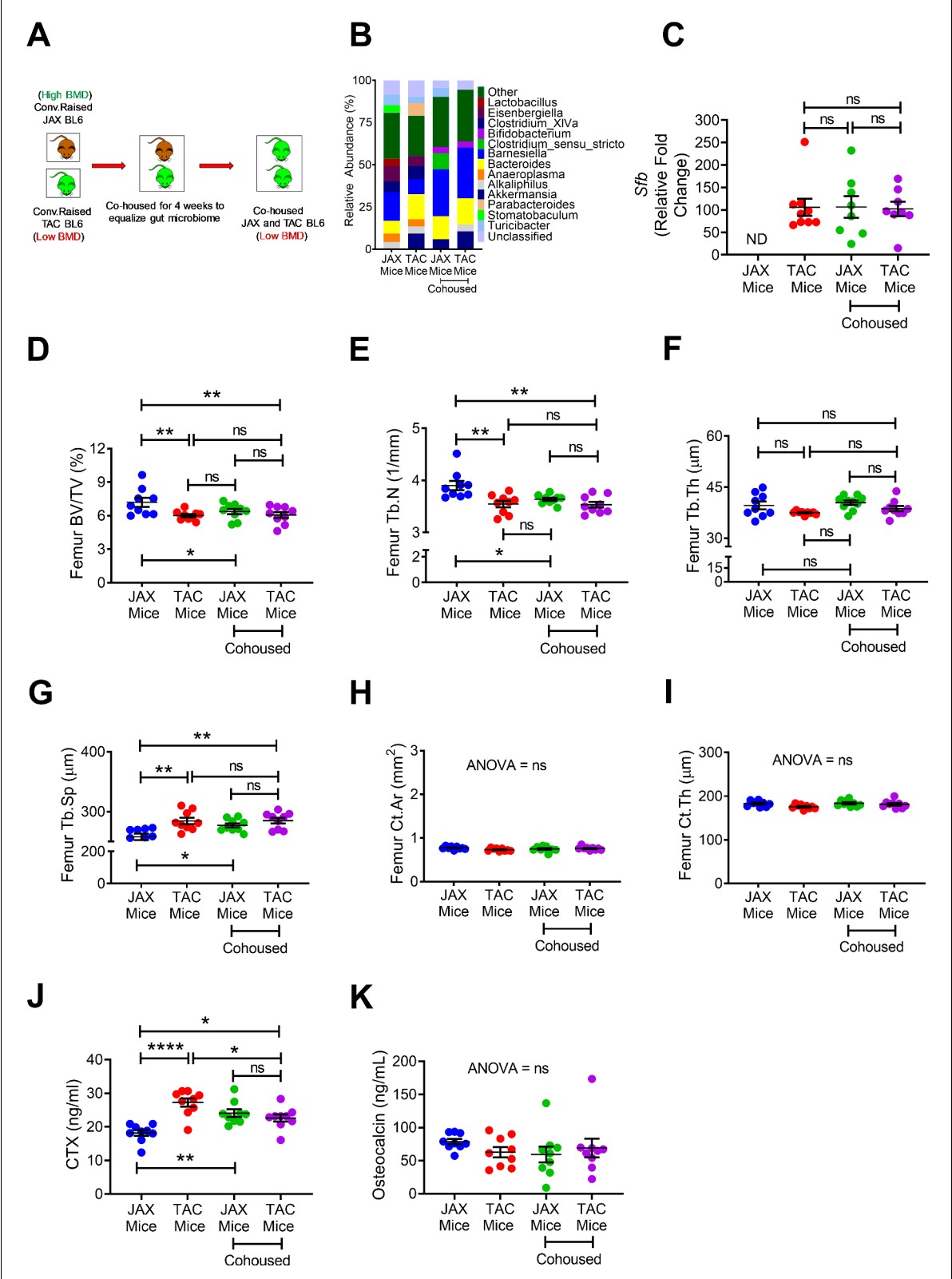

**Figure 3.** Effects of co-housing on bone volume, structure and turnover. (**A**) Diagram of the experimental outline. 10-week-old female JAX and TAC mice were housed separately, or co-housed for 4 weeks. (**B**) Effects of co-housing on the composition and frequency of microbiome. (**C**) Quantitative PCR analysis of *Sfb* and total bacterial 16S rRNA genes in fecal samples. (**D**) Femoral trabecular bone volume fraction (BV/TV). (**E**) Trabecular number (Tb.N). (**F**) Trabecular thickness (Tb.Th). (**G**) Trabecular separation (Tb.Sp). (**H**) Cortical Area (Ct.Ar). (**I**) Cortical thickness (Ct.Th). (**J**) Serum levels of CTX.
*Figure 3 continued on next page*

*Figure 3 continued*

(K) Serum levels of OCN. n = 8–9 mice per group. Data were expressed as mean ± SEM. All data were normally distributed according to the Shapiro-Wilk normality test. Data were analyzed by two-way ANOVA and post hoc tests applying the Bonferroni correction for multiple comparisons. *p<0.05, **p<0.01, ****p<0.0001 compared to indicated groups.

The online version of this article includes the following source data for figure 3:

**Source data 1.**

Our observations were obtained using the powerful model of SFB-induced bone loss (*Yu et al., 2020*; *Hathaway-Schrader et al., 2020*), which allowed us to faithfully measure microbiome-dependent osteo-immunological events.

These observations have considerable societal implications, as they provide rational to support the feasibility of pre-, pro- and post-biotic interventions (*Zaiss et al., 2019*) to enhance post-natal bone development. Because we show that the offspring inherits the microbiome from the mother, these interventions may include supplementing the microbiome of gestating females with bone anabolic bacterial strains which may then be transferred to offspring at birth. In addition, our observations may support the rational of directly supplementing the neonate with beneficial bacteria at birth. Whether introduced strains establish permanent residency is an open question, especially due to the reported change and establishment of the adult gut microbiome by 2 to 3 years of age (*Yatsunenko et al., 2012*; *Rodríguez et al., 2015*; *Kashtanova et al., 2016*). In the event that permanent residency is not established, then further bacteriotherapy interventions of infants at 2 to 3 years of age may be necessary. Discovering the parameters whereupon beneficial strains permanently engrafted into the microbiome throughout adulthood will be essential endeavors in efforts to treat microbial impediments to optimal post-natal bone development.

The transfer of phenotypes within the microbiome by cohabitation has long intrigued investigators. Contributing environmental factors that shape our microbial communities include diet, age, our surroundings, and the individuals with whom we interact (*Benson et al., 2010*). Furthermore, contact with pets, especially dogs, are influencing factors (*Song et al., 2013*). Because mice are coprophagic, they provide a robust model to establish the efficiency of microbiome transmission between animals, and importantly also allows the detection of microbiome-induced phenotypes (*Robertson et al., 2019*; *Caruso et al., 2019*). Our data supported the notion that the gut microbiome is a factor of non-genomic transmission of bone structure and bone turnover between animals, since acquisition of an SFB$^+$ microbiome by co-housing lowered indices of post-natal bone development in mice irrespective of genetic strain. Clearly, the extent to which these observations are applicable to humans depend on the surroundings, including sanitation practices within the home and the efficiency of waste treatment within the municipalities. While in healthy mice Th17 cells are induced mostly by SFB, in healthy humans there are about 20 non-virulent gut bacterial strains known to induce Th17 cell differentiation (*Atarashi et al., 2015*; *Tan et al., 2016*). Among them the most are Bifidobacterium adolescentis, Staphylococcus saprophyticus, Klebsiella, *Enterococcus faecalis* and Acinetobacter baumanii (*Atarashi et al., 2015*; *Tan et al., 2016*). Since these strains are not equally potent, the presence of one or more of Th17 cell-inducing bacteria and their relative frequency is likely to affect skeletal development. Thus, identifying a source and then implementing evasion strategies to prevent the acquisition of microbes that induce Th17 cells in the gut may be a potential approach to avoid impediments to optimal post-natal bone development.

Despite reports that the gut microbiota of healthy people shows resilience and a capacity for recovery following antibiotic treatment (*Palleja et al., 2018*), we showed that FMT immediately after antibiotic treatment potently modified post-natal bone development phenotype in mice. Our observations are evidence that following a course of broad-spectrum of antibiotics, the microbiome may indeed be modified by FMT to introduce bacterial strains that may elicit modulatory effects on bone health and disease. Whether the newly introduced bacterial strains within the FMT may become permanent residents over the lifetime of the organism is yet to be determined. However, in our approach using mice, we show that 4 weeks colonization of microbiome materials introduced by FMT was sufficient to elicit a phenotype. This period required for microbes to elicit their associated phenotypes may vary and likely to depend on the potency whereby a given bacterial strain elicits its influence. Nevertheless, since the lifespans of humans are considerably longer than experimental science timeframes, which was up to 8 weeks in the models outlined in this manuscript, the acquisition

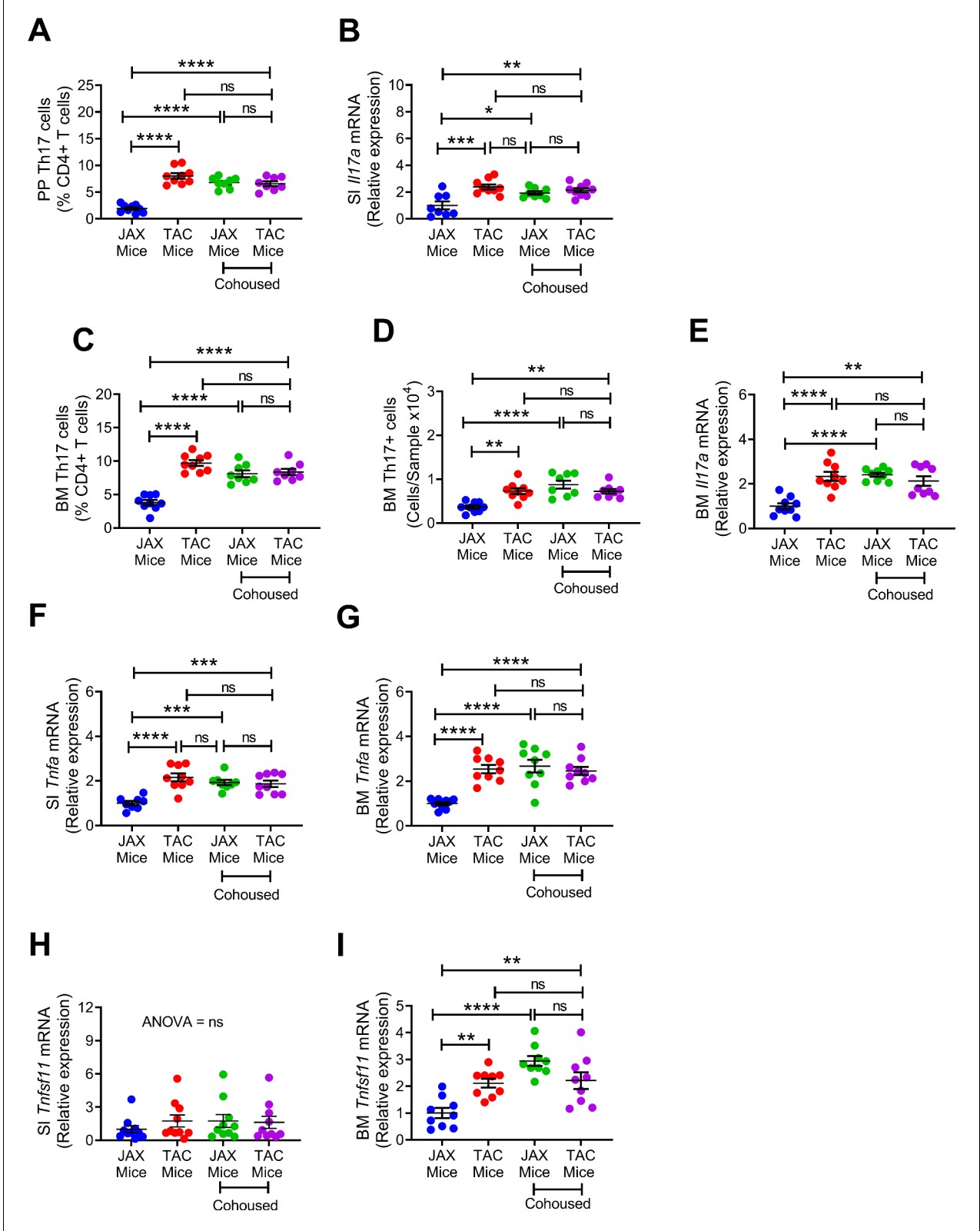

**Figure 4.** Effects of cohousing on the frequency of BM and PP Th17 cells and transcript levels of inflammatory cytokines. (**A**) Relative frequency of Th17 cells (IL-17+ CD4+ T cells) in PP. (**B**) SI *Il17a* mRNA levels. (**C,D**) Relative and absolute frequency of BM Th17 cells. (**E**) BM *Il17a* mRNA levels. (**F**) SI *Tnfa* (TNF) mRNA levels. (**G**) BM *Tnfa* mRNA levels. (**H**) SI *Tnfsf11* mRNA levels. (**I**) BM *Tnfsf11* mRNA levels. n = 8–9 mice per group. Data were expressed as

*Figure 4 continued on next page*

*Figure 4 continued*

mean ± SEM. All data were normally distributed according to the Shapiro-Wilk normality test. Data were analyzed by 2-way ANOVA and post hoc tests applying the Bonferroni correction for multiple comparisons. *p<0.05, **p<0.01, ***p<0.001, ****p<0.0001 compared to indicated groups.

The online version of this article includes the following source data for figure 4:

**Source data 1.**

and establishment of a Th17 cell expanding microbe early in life may have pernicious effects on post-natal bone metabolism over an entire lifetime.

An interesting finding of our studies was that increased expression of *Il17a* in the gut and the BM was associated with increased expression of *Tnfsf11* in the BM but not in the gut. IL-17 is known to induce TNFSF11 production by osteoblasts and osteocytes (*Lee, 2013*; *Li, 2018*). Therefore, it is likely that osteoblasts and osteocytes might be the source of TNFSF11 in the BM of mice in response to elevated levels of BM IL-17, and that the absence of these lineages and other TNFSF11 producing cells sensitive to IL-17 in the intestine may explain the inability of IL-17 to stimulate TNFSF11 production in the gut.

It should also be noted that the effects of the microbiome on skeletal development were more potent in trabecular bone than in cortical bone. This is consistent with previous reports where we found the microbiome not to regulate cortical bone (*Li et al., 2016*; *Yu et al., 2020*; *Li et al., 2020*; *Tyagi et al., 2018*). This is likely to reflect the fact that the microbiome regulates bone acquisition mostly via T cells. These cells reside in the BM, where they have physical access to trabecular bone cells. In addition, cytokines secreted by T cells are present in relative high concentration in the BM, where they can affect bone cells and their precursors. By contrast, T cells have minimal physical access to cortical bone.

In each of the three experimental models utilized in the current study, the microbiome regulated skeletal development by modulating bone resorption without affecting bone formation. The reason for this phenomenon remains unknown, although it is tempting to speculate that the capacity of IL-17 to blunt bone formation (*Kim et al., 2014*) might have been offset by gut microbes that stimulate bone formation such as bacteria that generate short chain fatty acids (*Zaiss et al., 2019*).

In summary, our data showed that the gut microbiome is a non-genomic heritable factor that contributes to the variance of bone volume, structure and turnover within populations. The specific effects of the microbiome on bone depend on its diversity and composition. The presence in the gut microbiome of saprophytic strains capable of expanding intestinal Th17 cell in healthy individuals it is likely to a predictor of suboptimal skeletal development and low peak bone mass. Furthermore, the findings of this study have significant implications to the practice of FMT from healthy donors to recipients, where the transfer of microbes that trigger the expansion of gut Th17 cells could lead to lower bone density in recipients. On the other hand, we also identified the method or timing of microbiome transfer as a window of opportunity to apply bacteriotherapeutic interventions to overcome suboptimal skeletal maturation.

# Materials and methods

## Key resources table

| Reagent type (species) or resource | Designation | Source or reference | Identifiers | Additional information |
|---|---|---|---|---|
| strain, strain background (*M. musculus*) | C57BL/6J | Jackson Laboratory | Cat# JAX:000664, RRID:IMSR_JAX:000664 | Mice for in vivo experiments |
| strain, strain background (*M. musculus*) | C57BL/6NTac | Taconic biosciences | Cat# TAC:b6, RRID:IMSR_TAC:b6 | Mice for in vivo experiments |
| strain, strain background (*M. musculus*) | C3H/HeNTac | Taconic biosciences | Cat# TAC:c3h, RRID:IMSR_TAC:c3h | Mice for in vivo experiments |

*Continued on next page*

Continued

| Reagent type (species) or resource | Designation | Source or reference | Identifiers | Additional information |
|---|---|---|---|---|
| Chemical compound, drug | Ampicillin | Sigma-Aldrich | Cat# A9393 | For Microbiota depletion |
| Chemical compound, drug | Vancomycin | Selleckchem | Cat# S2575 | For Microbiota depletion |
| Chemical compound, drug | Neomycin Sulfate | Sigma-Aldrich | Cat# 1062540100 | For Microbiota depletion |
| Chemical compound, drug | Metronidazole | Sigma-Aldrich | Cat# M3761 | For Microbiota depletion |
| Chemical compound, drug | Trizol | Life Technologies | Cat# 15596018 | Cells Sample collection for total RNA isolation |
| Chemical compound, drug | Zombie NIR | Biolegend | Cat# 423105 | FACS (0.1 ul per test) |
| Chemical compound, drug | Cell Activation Cocktail (without Brefeldin A) | Biolegend | Cat# 423301 | Cell Activation Cocktail |
| Chemical compound, drug | Monensin Solution | Biolegend | Cat# 420701 | Monensin for cell activation |
| Chemical compound, drug | Fixation Buffer | Invitrogen | Cat# 2178648 | For cell fixation |
| Chemical compound, drug | Permeabilization Buffer | Invitrogen | Cat# 2229113 | Cell Permeabilization Buffer |
| Antibody | Anti-Mouse CD16/32 (clone 93) (Mouse monoclonal) | Biolegend | Cat# 101302, RRID:AB_312801 | FACS (1 ul per test) |
| Antibody | BV 510-CD45 (clone 30-F11) (Mouse monoclonal) | Biolegend | Cat# 103138, RRID:AB_2563061 | FACS (1 ul per test) |
| Antibody | BV 421-TCRβ (clone H57-597) (Mouse monoclonal) | Biolegend | Cat# 109230, RRID:AB_2562562 | FACS (1 ul per test) |
| Antibody | AF 700-CD3 (clone 17A2) (Mouse monoclonal) | Biolegend | Cat# 100216, RRID:AB_493697 | FACS (1 ul per test) |
| Antibody | PerCP/Cy5.5-CD4 (clone RM4-5) (Mouse monoclonal) | Biolegend | Cat# 100540, RRID:AB_893326 | FACS (1 ul per test) |
| Antibody | BV 711-CD8 (clone 53–6.7) (Mouse monoclonal) | BD Biosciences | Cat# 100748, RRID:AB_2562100 | FACS (1 ul per test) |
| Antibody | Anti-mouse PE-IL-17A (clone eBio17B7) (Mouse monoclonal) | Thermo Fisher Scientific | Cat# 12-7177-81, RRID:AB_763582 | FACS (1 ul per test) |
| sequenced-based reagent | cDNA Synthesis kit | Invitrogen | Cat# 18080–051 | For RT-PCR |
| sequenced-based reagent | SYBR GREEN | Applied Biosciences | Cat# 4367659 | For RT-PCR |

*Continued*

| Reagent type (species) or resource | Designation | Source or reference | Identifiers | Additional information |
|---|---|---|---|---|
| commercial assay or kit | Osteocalcin | Immunodiagnostic systems Ltd. | Cat# AC-12F1 | For the measurement of serum levels of Osteocalcin |
| commercial assay or kit | CTX | Immunodiagnostic systems Ltd. | Cat# AC-06F1 | For the measurement of serum levels of CTX |
| commercial assay or kit | DNA Stool Qiagen kit | Qiagen | Cat# 51604 | For the 16S and SFB PCR |
| software, algorithm | Flow cytometry | LSR II system | BD Biosciences | |
| software, algorithm | FlowJo software | Tree Star, Inc | NA | |
| software, algorithm | MicroCT-40 scanner | Scanco | NA | |
| software, algorithm | QIIME | Open Source | NA | |
| software, algorithm | GraphPad Prism 8 | GraphPad Software | NA | |

All in vivo experiments were carried out in female mice. All conventionally raised mice entering Emory University were shipped to the same room in the same vivarium within the Whitehead Biomedical Research Building. All conventionally raised mice were maintained under a general housing environment and fed sterilized food (5V5R chow) and autoclaved water ad libitum. GF pregnant dam mice used for the FMT experiment were housed in a Tecniplast ISOcage P - Bioexclusion System within the Emory Gnotobiotic Animal Core. All mice were acclimatized within our facility for 3 days before experimentation.

## FMT in GF dams

Conv. R C57BL/6 and C3H/HeN mice and GF C3H/HeN mice were purchased from Taconic biosciences (Rensselaer, NY). Fecal material from 16 week-old Conv.R C57BL/6 and C3H/HeN mice was transferred into GF pregnant C57BL/6 and C3H/HeN mice of the opposite genotype (or same genotype to generate controls) through oral gavage on alternate days between E4 and E15 as described (*Gomez de Agüero et al., 2016*). Pregnancies were timed according to the day of vaginal plug. Fecal material donor mice were housed in Tecniplast bio-exclusion ISO cages to maintain the original microbiota until 16 weeks of age. Pregnant mice and offspring were housed in hermetically sealed ISO bio-exclusion cages. Once weaned, the offspring was housed alone in bio-exclusion ISO cages until the pups reach 16 weeks of age. To control for host-microbiome reciprocal interactions, the microbiome of each group of donor mice were normalized by routinely exchanging bedding between cages.

## Co-Housing

For the co-housing experiment, 10-week-old female Conv.R C57BL/6 mice were purchased from TAC and JAX laboratories. All the mice were fed sterile water and chow. Control groups were housed separately for 4 weeks. Experimental groups were co-housed with mice purchased from the other vendor that is JAX mice were co-housed with TAC mice for 4 weeks. Since mice are coprophagic, co-housing caused equalization of gut microbiome. After 4 weeks of co-housing mice were sacrificed.

## FMT in conv. R. C57BL/6 and C3H/HeN mice

8-week-old female Conv.R C57BL/6 and C3H/HeN mice were purchased from Taconic. All mice were fed sterile water and 5V5R chow. Prior to FM, recipient mice were treated with broad- spectrum antibiotics (Abx) (1 mg/mL ampicillin, 0.5 mg/mL vancomycin, 1 mg/mL neomycin sulfate, 1 mg/mL metronidazole) in drinking water for 2 weeks starting at the age of 8 weeks. 24 hr after completion

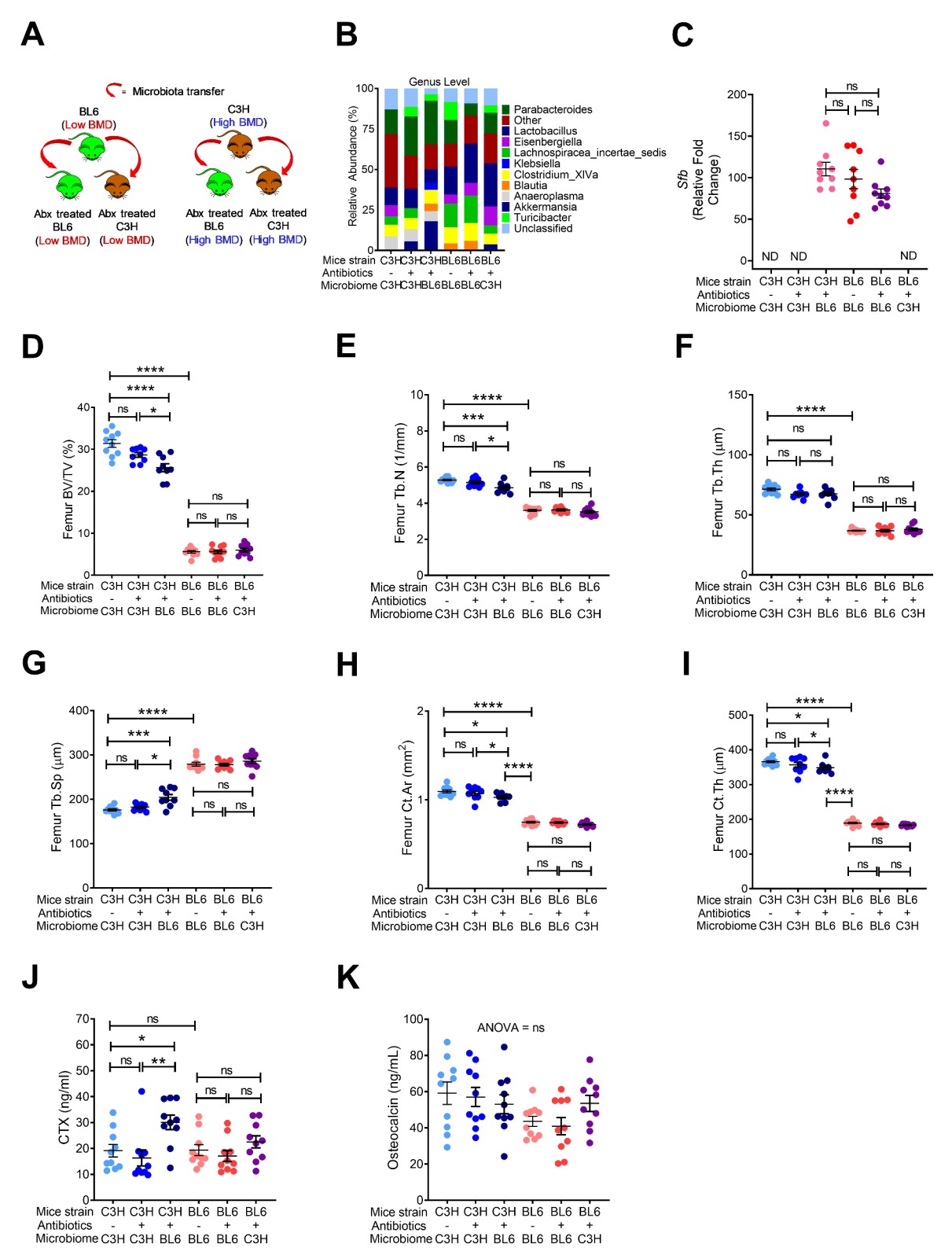

**Figure 5.** Effects of FMT on bone volume, structure and turnover. (**A**) Diagram of the experimental outline. (**B**) Fecal microbiome composition in C57BL/6 (BL6) mice and C3H/HeN (C3H) mice. (**C**) Quantitative PCR analysis of *Sfb* and total bacterial 16S rRNA genes in fecal samples. (**D**) Femoral trabecular bone volume fraction (BV/TV). (**E**) Trabecular number (Tb.N). (**F**) Trabecular thickness (Tb.Th). (**G**) Trabecular separation (Tb.Sp). (**H**) Cortical Area (Ct.Ar). (**I**) Cortical thickness (Ct.Th). (**J**) Serum levels of CTX. (**K**) Serum levels of OCN. n = 9–10 mice per group. Data were expressed as mean ± SEM. All data

*Figure 5 continued on next page*

*Figure 5 continued*

were normally distributed according to the Shapiro-Wilk normality test. Data were analyzed by 2-way ANOVA and post hoc tests applying the Bonferroni correction for multiple comparisons. *p<0.05, **p<0.01, ***p<0.001, ****p<0.0001 compared to indicated groups.

The online version of this article includes the following source data for figure 5:

**Source data 1.**

of the antibiotic treatment, a liquid suspension of fecal material was gavaged into recipient mice for three consecutive days using methods previously established in our laboratory (*Yu et al., 2020*; *Li et al., 2020*).

## Microbiota analysis

For microbiome analysis fecal pellets were collected from mice at sacrifice and DNA extracted from fecal samples using the MoBio DNA isolation kit. The V4 region of the 16S genes were amplified using the methods of *Caporaso et al., 2011*. Amplicons were sequenced on an Illumina MiSeq instrument at the Emory Integrated Genomics Core (EIGC). Analysis of the sequencing reads were done by the Emory Integrated Computational Core (EICC) using standard methodology for microbiome analysis. Briefly, the raw sequence was processed via QIIME, using closed-reference OTU picking and the Greengenes reference database. The resulting files were then moved into R and analyzed using the phyloseq package. Data processing involved demultiplexing, QC filtering (*Edgar et al., 2011*), OTU representation (*Edgar, 2010*), taxonomy assignment via a reference database (*Caporaso et al., 2010a*; *McDonald et al., 2012*; *Wang et al., 2007*) and phylogeny and diversity analysis (*Lozupone et al., 2007*) using the QIIME (*Caporaso et al., 2010b*) and MOTHUR (*Schloss et al., 2009*) pipelines.

## In-vitro μCT measurements

μCT scanning and analysis was performed as reported previously (*Tyagi et al., 2018*; *Grassi et al., 2016*; *Li et al., 2015*), using Scanco μCT-40 scanner. Voxel size was 12 μm$^3$ for in-vitro measurements of femur. For the femoral trabecular region, we analyzed 70 slices, beginning 50 slices below the distal growth plate. Femoral cortical bone was assessed using 80 continuous CT slides located at the femoral midshaft. X-ray tube potential was 70 kVp, 114 μA, and integration time was 200 ms for the in-vitro measurements. We used the thresholding approach described by *Bouxsein et al., 2010*. which is recommended by Scanco, the μCT-40 manufacturer, and involves a visual inspection and comparison of preview and slice-wise gray scale 2D images. The same threshold value was used for all measurements.

## Excision of Peyer's patches (PP) from small intestine

Peyer's patches (PP) cell isolation was performed as described (*Tyagi et al., 2018*; *Lefrancois and Lycke, 2001*). Briefly, the small intestine was removed and flushed of fecal content. PPs were excised and collected in 1 mL cooled RPMI1640. PPs were dissociated using the plunger of a 2.5 mL syringe and gently forced through a 70 μm cell strainer placed over a 50 mL tube. A single cell suspension was used for measuring the number of Th17 cells in PP by flow cytometry.

## Flow cytometry

Flow cytometry was performed on a LSR II system (BD Biosciences) and data were analyzed using FlowJo software (Tree Star, Inc, Ashland, OR). For cell surface staining: cells were stained with anti-mouse purified CD16/32 (clone 93), BV 510-CD45 (clone 30-F11), BV 421-TCRβ (clone H57-597), AF 700-CD3 (clone 17A2), PerCP/Cy5.5-CD4 (clone RM4-5) and BV 711-CD8 (clone 53–6.7) (BD Biosciences). The live cells were discriminated by Zombie NIR Fixable Viability Kit (Biolegend). For intracellular staining, cells were incubated with cell activation cocktail (Biolegend) in the presence of Monensin Solution at 37°C for 12 hr. Anti-mouse PE-IL-17A (clone eBio17B7) was added after cell fixation and permeabilization with Intracellular Fixation and Permeabilization Buffer Set (Thermo Fisher).

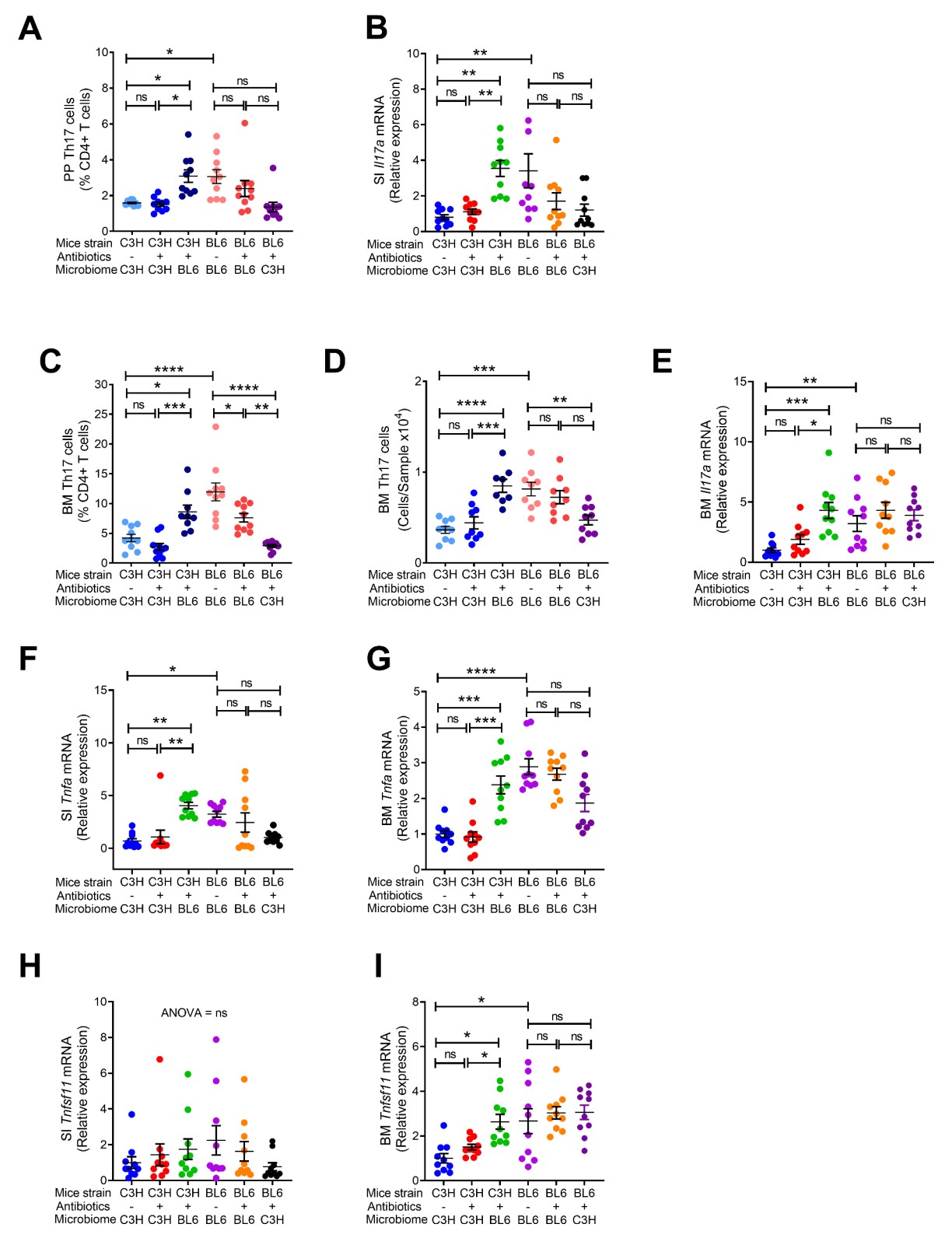

**Figure 6.** Effects of FMT on the frequency of BM and PP Th17 cells and transcript levels of inflammatory cytokines. (**A**) Relative frequency of Th17 cells (IL-17+ CD4+ T cells) in PP. (**B**) SI *Il17a* mRNA levels. (**C,D**) Relative and absolute frequency of BM Th17 cells. (**E**) BM *Il17a* mRNA levels. (**F**) SI *Tnfa* (TNF) mRNA levels. (**G**) BM *Tnfa* mRNA levels. (**H**) SI *Tnfsf11* mRNA levels. (**I**) BM *Tnfsf*11 mRNA levels. n = 9–10 mice per group. Data were expressed as

*Figure 6 continued on next page*

*Figure 6 continued*

mean ± SEM. All data were normally distributed according to the Shapiro-Wilk normality test. Data were analyzed by 2-way ANOVA and post hoc tests applying the Bonferroni correction for multiple comparisons. *p<0.05, **p<0.01, ***p<0.001, ****p<0.0001 compared to indicated groups.

The online version of this article includes the following source data for figure 6:

**Source data 1.**

### Markers of bone turnover

Markers of bone turnover, Osteocalcin (OCN) and C-terminal cross-linked telopeptide (CTX) were measured in serum by ELISA kit (Immunodiagnostic Systems Ltd. (Boldon, UK)).

### DNA extraction from fecal samples

Stool samples were collected directly into sterile tubes from live animals and snap-frozen before DNA extraction. Relative bacteria gene copies of Conv.R mice and antibiotic treated mice were confirmed at the end of the treatment period by fecal DNA extraction (DNA Stool Qiagen kit) as described in manufacture's protocol and subsequent qPCR using 515F (GTGCCAGCMGCCGCGG TAA) and 806R (GGACTACHVGGGTWTCTAAT) primers. Conv.R mice and water were used as positive and negative control, respectively.

### Quantitative real-time PCR

Total RNA was extracted from whole BM cells and small intestine using Trizol (Invitrogen). cDNA was synthesized from 1 mg total RNA with the superscript III first strand cDNA synthesis kit (Invitrogen). The mRNA expression levels of genes were analyzed by RT-PCR using an ABI Prism 7000 or One Step Plus Sequence Detection System and SYBR GREEN PCR Master Mix (Applied Biosystems, Foster City, CA, USA). Changes in relative gene expression between groups were calculated using the $2^{-\Delta\Delta Ct}$ method with normalization to 18S rRNA as previously described. All the primers used were designed by Primer Express Software v2.0 (Applied Biosystems) and most were validated in previous investigations (*Grassi et al., 2016*; *Li et al., 2015*). The primer sequences we used are as followed. 5'- ATTCGAACGTCTGCCCTATCA −3' (forward) and 5'- GTCACCCGTGGTCACCATG −3' (reverse) for 18 s rRNA. 5'- AACTCCAGGCGGTGCCT AT −3' (forward) and 5'- TGCCACAAGCAG-GAATG AGA −3' (reverse) for *Tnfa* mRNA. 5'- TGACGCCCACCTACAAC ATC −3' (forward) and 5'-CATCATGCAGTTCCGTCA GC −3' (reverse) for *Il17a* mRNA. 5'- 5'-CCTGATGAAAGGAGGGAGCA −3' (forward) and 5'- TGGAATTCAGAATTGCCCGA −3' (reverse) for *Tnfsf11* mRNA.

### Statistical analysis

All data are expressed as Mean ± SEM. Data were normally distributed according to the Shapiro-Wilk normality test. Data were analyzed by two-way ANOVA. This analysis included the main effects for animal strain and fecal matter transfer. When the statistical interaction was statistically significant (p<0.05) then tests were used to compare the differences between the treatment means for each animal strain, applying the Bonferroni correction for multiple comparisons.

### Study approval

All the animal procedures were approved by the Institutional Animal Care and Use Committee of Emory University.

## Acknowledgements

This study was supported by grants from the National Institutes of Health (DK112946, DK108842, and RR028009 to RP; DK098391 to RMJ).

# Additional information

## Funding

| Funder | Grant reference number | Author |
| --- | --- | --- |
| National Institutes of Health | DK112946 | Roberto Pacifici |
| National Institutes of Health | DK108842 | Roberto Pacifici |
| National Institutes of Health | RR028009 | Roberto Pacifici |
| National Institutes of Health | DK098391 | Rheinallt M Jones |

The funders had no role in study design, data collection and interpretation, or the decision to submit the work for publication.

## Author contributions

Abdul Malik Tyagi, Data curation, Formal analysis, Validation, Investigation, Visualization, Methodology, Writing - original draft, Writing - review and editing; Trevor M Darby, Emory Hsu, Subhashis Pal, Hamid Dar, Jonathan Adams, Formal analysis, Investigation, Methodology; Mingcan Yu, Data curation, Formal analysis, Investigation, Methodology; Jau-Yi Li, Data curation, Investigation, Visualization, Methodology; Rheinallt M Jones, Resources, Supervision, Investigation, Writing - review and editing; Roberto Pacifici, Conceptualization, Supervision, Funding acquisition, Investigation, Writing - original draft, Project administration, Writing - review and editing

## Author ORCIDs

Abdul Malik Tyagi (iD) https://orcid.org/0000-0001-9012-5779
Subhashis Pal (iD) https://orcid.org/0000-0002-3916-5545
Roberto Pacifici (iD) https://orcid.org/0000-0001-6077-8250

## Ethics

Animal experimentation: This study was performed in strict accordance with the recommendations in the Guide for the Care and Use of Laboratory Animals of the National Institutes of Health. All of the animals were handled according to approved institutional animal care and use committee (IACUC) protocols (201800029) of Emory University.

## Decision letter and Author response

Decision letter https://doi.org/10.7554/eLife.64237.sa1
Author response https://doi.org/10.7554/eLife.64237.sa2

# Additional files

## Supplementary files

• Transparent reporting form

## Data availability

All data generated or analyzed during this study are included in the manuscript and supporting files.

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
