## [Decision Letter]

**Acceptance summary:**

Your manuscript was seen as worthy and further illuminated a new area of biology by investigating the influence of gut microbiome on skeletal maturation in a manner that was inheritable. This leads to translational implications for improving skeletal health by manipulating the microbiome.

**Decision letter after peer review:**

Thank you for submitting your article "The gut microbiota is a transmissible determinant of skeletal maturation" for consideration by *eLife*. Your article has been reviewed by two peer reviewers. The reviewers have opted to remain anonymous.

The reviewers have discussed the reviews with one another and the Reviewing Editor has drafted this decision to help you prepare a revised submission.

The study is seen as novel and of general interest because it is the first to demonstrate that the microbiome is a non-genomic determinant of bone mass heritability. It includes novel approaches, such as the evaluation of the effects on the offspring of microbiome transfer in the mother. The analysis of bone indices has been conducted using state of the art methods, and the conclusions are well supported by the presented data. With that said issues below need to be addressed before the study is ready for publication in *eLife*.

1) SFB is a specific inducer of Th17 differentiation in mice. This study demonstrates the importance of SFB as regulator of postnatal bone mass acquisition in mice. Are these findings relevant in humans?

2) Please discuss how intestinal Th17 cells are induced in humans and if bacteria equivalent to SFB have been described in humans. RANKL is a critical inducer of osteoclast formation, and the manuscript shows that BM RANKL is regulated by the microbiome. Please discuss what is the cellular source of BM RANKL and whether the authors believe that RANKL is induced by IL-17.

3) The effects of the microbiome appear to be more potent in trabecular bone than in cortical bone. How is this explained? Is this consistent with previous reports about the effects of microbiome in bone?

4) Introduction: The authors refer to the role of GM as regulator of skeletal mass in humans making references to two papers (Wang et al., 2017; Li et al., 2019), these papers are not related to skeletal maturation, they refer to adults.

5) Materials and methods: Please briefly introduce the difference in BMD between the two mice strain used in order to help the reader's comprehension.

6) Results: No differences in bone formation markers in mice treated with different GM manipulation is reported. One could expect that the effect on skeletal maturation is mediated by bone modelling and bone formation. Moreover, a role for IL-17 in the inhibition of osteobastogenesis and of bone formation has been suggested. It is suggested to that a paragraph in the Discussion be included in order to clarify this point.

7) There are some typos as Payer's patches in the legend of supplemental figure.

---

## [Author Response]

1) SFB is a specific inducer of Th17 differentiation in mice. This study demonstrates the importance of SFB as regulator of postnatal bone mass acquisition in mice. Are these findings relevant in humans?

We believe that our findings are relevant in humans. In fact, while in healthy mice Th17 cells are induced mostly by SFB, in healthy humans, there are about 20 non-virulent gut bacterial strains known to induce Th17 cell differentiation (1, 2). Therefore, we hypothesize that skeletal growth in humans is influenced by the presence and the frequency of one or more of these Th17 cell-inducing bacteria. This point is now discussed in the revised manuscript where we state:

”While in healthy mice Th17 cells are induced mostly by SFB, in healthy humans there are about 20 non-virulent gut bacterial strains known to induce Th17 cell differentiation. Since these strains are not equally potent, the presence of one or more of Th17 cell-inducing bacteria and their relative frequency is likely to affect skeletal development. Thus, identifying a source and then implementing evasion strategies to prevent the acquisition of microbes that induce Th17 cells in the gut may be a potential approach to avoid impediments to optimal post-natal bone development”.

2) Please discuss how intestinal Th17 cells are induced in humans and if bacteria equivalent to SFB have been described in humans. RANKL is a critical inducer of osteoclast formation, and the manuscript shows that BM RANKL is regulated by the microbiome. Please discuss what is the cellular source of BM RANKL and whether the authors believe that RANKL is induced by IL-17.

In the revised manuscript we describe the role of SFB-like bacteria in the induction to Th17 cells in humans. Please see reply to point #1 above. We also discuss our hypothesis that RANKL is likely to be secreted by osteoblasts/osteocytes and possibly T cells in response to IL-17. Our hypothesis is supported by pertinent literature and by an earlier report from our laboratory in which we demonstrated that IL-17 signaling in osteoblasts/osteocytes stimulates RANKL production in concert with PTH (3). The following paragraph has been added to the Discussion:

“An interesting finding of our studies was that increased expression of IL-17 in the gut and the BM was associated with increased expression of RANKL in the BM but not in the gut. IL-17 is known to induce RANKL production by osteoblasts and osteocytes. Therefore, it is likely that osteoblasts and osteocytes might be the source of RANKL in the BM of mice in response to elevated levels of BM IL-17, and that the absence of these lineages and other RANKL producing cells sensitive to IL-17 in the intestine may explain the inability of IL-17 to stimulate RANKL production in the gut”.

3) The effects of the microbiome appear to be more potent in trabecular bone than in cortical bone. How is this explained? Is this consistent with previous reports about the effects of microbiome in bone?

This is a very interesting point that is been addressed by adding the following paragraph to the Discussion:

“It should also be noted that the effects of the microbiome on skeletal development were more potent in trabecular bone than in cortical bone. This is consistent with previous reports where we found the microbiome not to regulate cortical bone. This is likely to reflect the fact that the microbiome regulates bone acquisition mostly via T cells. These cells reside in the BM, where they have physical access to trabecular bone cells. In addition, cytokines secreted by T cells are present in relative high concentration in the BM, where they can affect bone cells and their precursors. By contrast, T cells have minimal physical access to cortical bone”.

4) Introduction: The authors refer to the role of GM as regulator of skeletal mass in humans making references to two papers (Wang et al., 2017; Li et al., 2019), These papers are not related to skeletal maturation, they refer to adults.

Point well taken. The references and the associated paragraph have been deleted.

5) Materials and methods: Please briefly introduce the difference in BMD between the two mice strain used in order to help the reader's comprehension.

Done.

6) Results: No differences in bone formation markers in mice treated with different GM manipulation is reported. One could expect that the effect on skeletal maturation is mediated by bone modelling and bone formation. Moreover, a role for IL-17 in the inhibition of osteobastogenesis and of bone formation has been suggested. It is suggested to that a paragraph in the Discussion be included in order to clarify this point.

This important point has been addressed by adding the following paragraph to the Discussion:

”In each of the three experimental models utilized in the current study, the microbiome regulated skeletal development by modulating bone resorption without affecting bone formation. The reason for this phenomenon remains unknown, although it is tempting to speculate that the capacity of IL-17 to blunt bone formation (4) might have been offset by gut microbes that stimulate bone formation such as bacteria that generate short chain fatty acids (5)”.

7) There are some typos as Payer's patches in the legend of supplemental figure.

We apologize for the typo, which has been fixed.

References

1) Atarashi K, Tanoue T, Ando M, Kamada N, Nagano Y, Narushima S, Suda W, Imaoka A, Setoyama H, Nagamori T, Ishikawa E, Shima T, Hara T, Kado S, Jinnohara T, Ohno H, Kondo T, Toyooka K, Watanabe E, Yokoyama S, Tokoro S, Mori H, Noguchi Y, Morita H, Ivanov, II, Sugiyama T, Nunez G, Camp JG, Hattori M, Umesaki Y, Honda K. Th17 Cell Induction by Adhesion of Microbes to Intestinal Epithelial Cells. Cell. 2015;163(2):367-80. Epub 2015/09/29. doi: 10.1016/j.cell.2015.08.058. PubMed PMID: 26411289; PMCID: PMC4765954.

2) Tan TG, Sefik E, Geva-Zatorsky N, Kua L, Naskar D, Teng F, Pasman L, Ortiz-Lopez A, Jupp R, Wu HJ, Kasper DL, Benoist C, Mathis D. Identifying species of symbiont bacteria from the human gut that, alone, can induce intestinal Th17 cells in mice. Proc Natl Acad Sci U S A. 2016;113(50):E8141-E50. Epub 2016/12/03. doi: 10.1073/pnas.1617460113. PubMed PMID: 27911839; PMCID: PMC5167147.

3) Li JY, Yu M, Tyagi AM, Vaccaro C, Hsu E, Adams J, Bellido T, Weitzmann MN, Pacifici R. IL-17 Receptor Signaling in Osteoblasts/Osteocytes Mediates PTH-Induced Bone Loss and Enhances Osteocytic RANKL Production. J Bone Miner Res. 2018. Epub 2018/11/07. doi: 10.1002/jbmr.3600. PubMed PMID: 30399207.

4) Kim YG, Park JW, Lee JM, Suh JY, Lee JK, Chang BS, μm HS, Kim JY, Lee Y. IL-17 inhibits osteoblast differentiation and bone regeneration in rat. Arch Oral Biol. 2014;59(9):897-905. Epub 2014/06/08. doi: 10.1016/j.archoralbio.2014.05.009. PubMed PMID: 24907519.

5) Zaiss MM, Jones RM, Schett G, Pacifici R. The gut-bone axis: how bacterial metabolites bridge the distance. J Clin Invest. 2019;130. Epub 2019/07/16. doi: 10.1172/JCI128521. PubMed PMID: 31305265.